# The association between cardiopulmonary exercise testing and postoperative outcomes in patients with lung cancer undergoing lung resection surgery: A systematic review and meta-analysis

Nabeela Arbee-Kalidas⊙*⊙, Hlamatsi Jacob Moutlana⊙, Yoshan Moodley⊙⊙, Moses Mogakolodi Kebalepile⊙, Palesa Motshabi Chakane⊙

Department of Anaesthesiology, Faculty of Health Sciences, University of the Witwatersrand, Johannesburg, South Africa

⊙ These authors contributed equally to this work.
* nabeela_arbee@yahoo.com

## Abstract

### Background

Exercise capacity should be determined in all patients undergoing lung resection for lung cancer surgery and cardiopulmonary exercise testing (CPET) remains the gold standard. The purpose of this study was to investigate associations between preoperative CPET and postoperative outcomes in patients undergoing lung resection surgery for lung cancer through a review of the existing literature.

### Methods

A search was conducted on PubMed, Scopus, Cochrane Library and CINAHL from inception until December 2022. Studies investigating associations between preoperative CPET and postoperative outcomes were included. Risk of bias was assessed using the QUIPS tool. A random effect model meta-analysis was performed. $I^2 > 40\%$ indicated a high level of heterogeneity.

### Results

Thirty-seven studies were included with 6450 patients. Twenty-eight studies had low risk of bias. $\dot{V}O_2$ peak is the oxygen consumption at peak exercise and serves as a marker of cardiopulmonary fitness. Higher estimates of $\dot{V}O_2$ peak, measured and as a percentagee of predicted, showed significant associations with a lower risk of mortality [MD: 3.66, 95% CI: 0.88; 6.43 and MD: 16.49, 95% CI: 6.92; 26.07] and fewer complications [MD: 2.06, 95% CI: 1.12; 3.00 and MD: 9.82, 95% CI: 5.88; 13.76]. Using a previously defined cutoff value of > 15mL/kg/min for $\dot{V}O_2$ peak, showed evidence of decreased odds of mortality [OR: 0.55, 95% CI: 0.28–0.81] and but not decreased odds of postoperative morbidity [OR: 0.82, 95%

**Data Availability Statement:** All relevant data are within the manuscript and its Supporting information files.

**Funding:** The author(s) received no specific funding for this work.

**Competing interests:** The authors have declared that no competing interests exist.

**Abbreviations:** AT, Anaerobic threshold; ATP, Adenosine triphosphate; BTS, British Thoracic Society; CINAHL, Cumulative Index to Nursing and Allied Health Literature; CPET, Cardiopulmonary exercise testing (CPET); MD, Mean difference; MeSH, Medical Subject Headings; n, Number of patients; NSCLC, Non-small cell lung cancer; POETTS, Perioperative Exercise Testing and Training Society; PRISMA, Preferred Reporting Items for Systematic reviews and Meta-Analysis; QUIPS, Quality in Prognosis Studies; RoB, Risk of bias; SCTS, Society of Cardiothoracic Surgeons of Great Britain and Ireland; SD, Standard deviation; $\dot{V}E/\dot{V}CO_2$, Ventilatory equivalents; $\dot{V}O_2$ peak, Oxygen consumption at peak exercise.

CI: 0.64–1.00]. There was no relationship between $\dot{V}E/\dot{V}CO_2$ slope, which depicts ventilatory efficiency, with mortality [MD: -9.60, 95% CI: -27.74; 8.54] however, patients without postoperative complications had a lower preoperative $\dot{V}E/\dot{V}CO_2$ [MD: -2.36, 95% CI: -3.01; -1.71]. Exercise load and anaerobic threshold did not correlate with morbidity or mortality. There was significant heterogeneity between studies.

## Conclusions

Estimates of cardiopulmonary fitness as evidenced by higher $\dot{V}O_2$ peak, measured and as a percentage of predicted, were associated with decreased morbidity and mortality. A cutoff value of $\dot{V}O_2$ peak > 15mL/kg/min was consistent with improved survival but not with fewer complications. Ventilatory efficiency was associated with decreased postoperative morbidity but not with improved survival. The heterogeneity in literature could be remedied with large scale, prospective, blinded, standardised research to improve preoperative risk stratification in patients with lung cancer scheduled for lung resection surgery.

## Introduction

Surgical resection is the first choice of treatment for stage I and stage II non-small cell lung cancers (NSCLC) with an improved 5-year survival rate from 11.3% to 44.9% [1]. For patients to benefit from the advantages of surgery, meticulous preoperative risk stratification needs to be performed, especially in this population who are generally elderly and have pre-existing cardiopulmonary limitations. An important aspect of preoperative risk stratification is determining functional capacity through exercise testing.

Cardiopulmonary exercise testing (CPET) is a non-invasive and objective means of assessing cellular, circulatory and cardiopulmonary functions under metabolic stress. It remains the gold standard in determining a patient's functional capacity and "fitness for surgery" [2, 3]. Previous systematic reviews that investigated the utility of CPET in patients undergoing intra-abdominal surgeries [4], vascular surgeries [5], and other non-cardiopulmonary surgeries [6, 7] all found that better preoperative exercise performance is associated with improved postoperative outcomes. The utilisation of CPET is limited by the need for expensive equipment and trained personnel; the potential for incorrect display of graphical data; the over-reliance on individual threshold values for preoperative risk stratification and the lack of large-scale; and prospective, blinded and standardised literature [8, 9]. These limitations have precluded its use in every institution and for every patient.

Historically, an anaerobic threshold (AT) of < 11ml/kg/min was the primary CPET marker used to recognise high risk patients [10]. However, as the research has evolved, different predictors for determining high perioperative risk for different surgeries such as oxygen consumption at peak exercise ($\dot{V}O_2$ peak) and ventilatory equivalents ($\dot{V}E/\dot{V}CO_2$) have been identified. Benzo et al. investigated associations between $\dot{V}O_2$ peak and specific pulmonary complications following lung resection surgery in a meta-analysis [11]. They found that a mean $\dot{V}O_2$ peak of < 15mL/kg/min was associated with a higher risk of postoperative pulmonary complications. The limitation of their study was that they also included patients who underwent lung resection surgery for benign conditions. Their observed effects may therefore not be fully applicable to lung cancer patients. The cardiopulmonary limitations in patients with lung cancer could be attributed to the multisystemic effects of the malignancy such as

global inflammation, sarcopenia, anaemia, fatigue and the side effects of chemotherapy or radiation therapy which patients with benign conditions may not experience [12, 13].

The current British Thoracic Society (BTS) and the Society of Cardiothoracic Surgeons of Great Britain and Ireland (SCTS) guidelines recommended a $\dot{V}O_2$ peak value that is in agreement with Benzo et al [14]. They consider patients with a $\dot{V}O_2$ peak < 15mL/kg/min as high risk and recommend limited resection, radiotherapy or chemotherapy [14]. However, there is no exposition on the utility of other CPET variables such as AT or $\dot{V}E/\dot{V}CO_2$ in the current guidelines, and $\dot{V}O_2$ peak in relation to predicted values for the patients.

The understanding of CPET and the research surrounding CPET is developing quickly. For lung resection surgery, emphasis had always been on $\dot{V}O_2$ peak. Newer evidence since the last systematic review has emerged showing $\dot{V}E/\dot{V}CO_2$ to be independently predictive of mortality [15]. Potentially, the use of other variables such as $\dot{V}E/\dot{V}CO_2$ or AT, singly or in combination, could have better predictive value for poor outcomes or be useful in prehabilitation in patients undergoing lung resection surgery. The current study sought to identify and pool data on the significance of these parameters in predicting adverse outcomes following lung resection surgery. This would serve as an indicator for further research in developing multi-marker predictive models.

## Objectives

The objective of this study was to investigate potential associations between preoperative CPET parameters and postoperative outcomes in patients undergoing lung resection surgery for lung cancer through a systematic review.

## Methods

### Protocol and registration

The study protocol was registered with PROSPERO (CRD42021259987), and adhered to the Preferred Reporting Items for Systematic reviews and Meta-Analysis (PRISMA) guidelines [16] (S1 Table).

### Ethics

An ethics waiver was obtained from the University of the Witwatersrand's Medical Human Research Ethics Committee (Reference number: W-CBP-210615-04).

### Eligibility criteria

The inclusion criteria for this review were as follows: 1) prognostic studies in which adult patients underwent lung resection surgery for lung cancer; 2) preoperative CPET was performed; 3) postoperative outcomes (mortality and morbidity) were measured; 4) a comparison of preoperative CPET variables with postoperative outcomes was done. The exclusion criteria were as follows: 1) thoracic surgery unrelated to lung cancer; 2) studies that included preoperative exercise programs; 3) systematic reviews, meta-analyses, case studies, letters to the editor and data from literature which would not yield sufficient information for inclusion and 4) studies not available in English full-text.

### Data sources

PubMed, Scopus, Cochrane Library and CINAHL databases were searched from 1967 until 22 December 2022. Medical Subject Heading (MeSH) terms and different keywords for

"preoperative," "cardiopulmonary exercise test," "lung cancer", "lung resection" and "postoperative outcomes" were used in the search (S2 Table). The search was augmented by reviewing the reference lists of retrieved articles. Limits were human and adult subjects.

## Data collection

Screening of titles and abstracts of potentially eligible studies was performed by two authors (NA & JM) with full-text articles assessed against the inclusion and exclusion criteria. Disagreements were resolved through discussion with a third reviewer (PM). The results from the database search were exported to EndNote 20 (Clarivate.™ Philadelphia, USA, 2022). Full text articles were sought using the University of the Witwatersrand's databases and interlibrary loans.

## Data extraction

Data extraction was performed by two authors independently (NA & JM) and further reviewed by two authors (PMC & MK) to ensure all data were accurate and complete.

The following data were extracted from eligible articles:

- General study details: author, year of publication and general conclusion

- Participants: sample size, patient selection, gender and surgery performed (pneumonectomy, lobectomy, wedge resection, segmentectomy, other such as thoracoscopy)

- CPET: type of CPET performed, if CPET was used in preoperative risk stratification i.e. blinding and all CPET variables used in the correlation analyses for each paper ($\dot{V}O_2$ peak measured and percentage of predicted, $\dot{V}E/\dot{V}CO_2$ AT, peak achieved exercise load)

- Outcomes: postoperative morbidity (in-hospital, 30-day, 60-day) and mortality (in-hospital, 30-day, 60-day, 90-day, $\geq$ 2 years)

## Risk of bias (quality) assessment

The quality of the studies' methods was assessed using the Quality in Prognosis Studies (QUIPS) tool which is recommended by the Cochrane Prognosis Methods Group [17]. Two authors (NA & JM) independently assessed the risk of bias (RoB) in each study. Disagreements were resolved through discussion and no mediation by a third party was required.

The QUIPS tool examined the RoB in observational prognostic studies in six areas, namely: study participation, attrition, prognostic factor measurement, outcome measurement, study confounding, and statistical analysis and reporting. This risk was expressed as a three-grade scale (high, moderate or low). All studies, irrespective of RoB grade, were included.

## Summary measures and synthesis of results

For the analysis, the studies were grouped based on outcomes (in-hospital, 30-day, and 60-day morbidity and in-hospital, 30-day, 60-day, 90-day and $\geq$ 2-year mortality) (S3 and S4 Tables). Studies presenting high variability of data type or format were analysed descriptively. The statistical analyses were conducted using Review Manager Version 5.4 (Copenhagen: The Nordic Cochrane Centre, The Cochrane Collaboration, 2020). All data were continuous and therefore the analysis was performed using the standardised mean difference (MD) of estimates. The degree of heterogeneity was measured using an $I^2$ test, with $I^2 > 40\%$ indicating a high level of heterogeneity. Additionally, a Tau score and Chi square statistic were used to assess the homogeneity of the studies, within a grouped outcome and between the groups of outcomes. Forest

plots were prepared for relevant outcomes. R Statistical package (version 4.2.2), visualized in RStudio (version 2023.02.3) were used to compute Forest plots. The pooled summary estimates (MD), with the accompanying 95% confidence intervals calculated were reported. The meta-analysis used a random effect method, and the SDs of the individual studies with their sample sizes were used to give every study a weighting. The common-effect assumption often required to perform a fixed-effect analysis, was not established, and therefore the acceptable alternative to a method implying an inverse-variance, was the random effect model. Statistical significance was set at $p < 0.05$.

## Results

### Study selection

The detailed process of the search is shown in the PRISMA flow diagram (Fig 1). The search strategy identified 1104 articles. Screening the reference lists yielded 10 additional articles. After removing duplicates, 1096 articles were screened. From the titles and abstracts, 1041 were excluded for not meeting the inclusion criteria. A total of 55 studies warranted a full-text review however 8 studies had no full English text. Of the 47 full English texts found, 10 studies were further excluded after full-text review for the following reasons: patients in the study underwent lung resection for non-malignant conditions (n = 7) [18–25] and studies did not compare CPET to outcomes (n = 3) [26–28]. This systematic review comprised 37 eligible studies [15, 24, 29–63]. Six studies were not included in the meta-analysis but were described as they were presented in the form of categories with no access to raw data and an inability to pool the data.

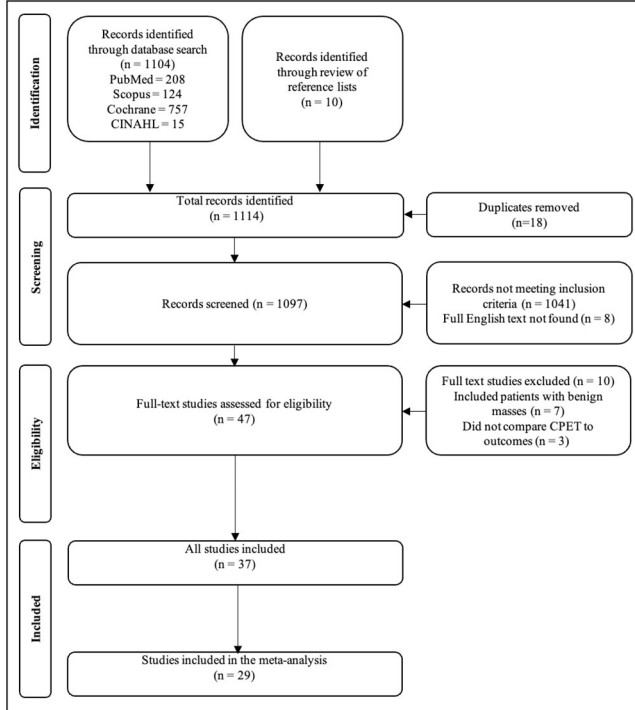

**Fig 1. PRISMA flow diagram for study retrieval.**

### Risk of bias assessment

Twenty-eight studies were assessed as having a low RoB, five studies had a moderate RoB, and four studies had a high RoB (Table 1). An incomplete description of study confounders was the most common cause of bias.

### General study and patient characteristics

In total, 6450 patients were included in the systematic review (Table 2). The pooled mean (SD) age was 64.70 (7.58) years. The sample sizes ranged from 12–1684 patients. Not all studies mentioned sex distribution, but for those that did, 3055 (71.41%) were male, and 1223 (28.59%) were female. Lobectomy was performed in 4213 (73.38%) patients, pneumonectomy in 721 (12.56%), a wedge resection in 365 (6.36%), a segmentectomy in 90 (1.57%), and an unspecified lung resection in 352 (6.13%) patients.

### CPET characteristics

CPET was performed on a cycle ergometer in 30 studies, a treadmill in six studies, and no clear mode of CPET was described in one study. Fifteen studies sent patients for CPET if they were considered high risk as per self-defined criteria or guidelines, and 22 sent all patients for CPET.

### Meta-analysis of outcome measures

**Mortality.** *The relationship between $\dot{V}O_2$ peak and mortality.* Overall, estimates of higher $\dot{V}O_2$ peak (mL/kg/min) showed significant associations with decreased mortality [MD: 3.66, 95% CI: 0.88; 6.43] however, there was significant heterogeneity ($I^2$ = 87%, $X^2$ = 45, df = 6, p < 0.01) (Fig 2). The subgroup analyses by time frame did not favour better survival (30-, 60- and 90-days).

*The relationship between $\dot{V}O_2$ peak as a percentage of predicted and mortality.* The total pooled estimates for $\dot{V}O_2$ peak as a percentage of predicted revealed that a higher mean value was associated with lower risk of mortality overall [MD: 16.49, 95% CI: 6.92; 26.07], and the subgroup analysis by time frame was similar (Fig 3).

*The relationship between $\dot{V}O_2$ peak > 15mL/kg/min and mortality.* $\dot{V}O_2$ peak > 15mL/kg/min was associated with decreased mortality in pooled data from 3 studies [OR: 0.55, 95% CI: 0.28–0.81] (Fig 4).

*The relationship between $\dot{V}E/\dot{V}CO_2$ and mortality.* The total pooled estimates of ventilatory efficiency emanated from a single study [15]. There was no relationship between $\dot{V}E/\dot{V}CO_2$ slope and $\dot{V}E/\dot{V}CO_2$ as a percentage of predicted with mortality [MD: -95.84, 95% CI: -27.77; 16.09] and [MD: -19.25, 95% CI: -122.01; 83.51] respectively (Fig 5).

*The relationship between exercise load and mortality.* There was no difference in the mean estimates of peak exercise load achieved (watts) between patients who died and those who survived postoperatively [MD: 20.76, 95% CI: -142.34; 183.86] (Fig 6). The studies displayed moderate heterogeneity ($I^2$ = 50%, $X^2$ = 2.01, df = 1, p = 0.16).

**Morbidity.** *The relationship between $\dot{V}O_2$ peak and morbidity.* Our pooled analysis demonstrated that patients without complications presented for surgery with a higher $\dot{V}O_2$ peak (mL/kg/min) compared to patients with postoperative complications [MD: 2.06, 95% CI: 1.12; 3.00] however, the data was significantly heterogeneous ($I^2$ = 93%, $X^2$ = 313.83, df = 23, p < 0.01). (Fig 7). A sub-analysis of $\dot{V}O_2$ peak (mL/kg/min) at 30-days did not show an association with better outcomes. Patients with higher $\dot{V}O_2$ peak (L/min) had better overall

**Table 1. QUIPS risk of bias assessment.**

| Author | Study participation | Study attrition | Prognostic factor management | Outcome measurement | Study confounding | Statistical analysis and reporting | Overall |
|---|---|---|---|---|---|---|---|
| Bechard et al. (1987) | Low | Low | Low | Low | Moderate | Low | Low |
| Begum et al. (2016) | Low | Moderate | Moderate | Low | Moderate | Low | High |
| Berggren et al. (1984) | Low | Low | Moderate | Low | Moderate | Low | Moderate |
| Brat et al. (2016) | Low | Moderate | Low | Low | Low | Low | Low |
| Brunelli et al. (2009) | Moderate | Low | Low | Low | Low | Low | Low |
| Brutsche et al. (2000) | Low | Moderate | Low | Low | Low | Low | Low |
| Campione et al. (2010) | Low | Low | Low | Low | Low | Low | Low |
| Chouinard et al. (2022) | Low | Low | Low | Low | Low | Low | Low |
| Colman et al. (1982) | Moderate | Low | Low | Low | Low | Low | Low |
| Epstein et al. (1993) | Low | Low | Low | Low | Moderate | Low | Low |
| Fang et al. (2013) | Low | Low | Moderate | Moderate | Moderate | Moderate | High |
| Fajardo et al. (2014) | Low | Moderate | Low | Low | Low | Low | Low |
| Holden et al. (1992) | Low | Low | Low | Low | Moderate | Low | Low |
| Kasikcioglu et al. (2009) | Moderate | Low | Low | Low | Low | Low | Low |
| Larsen et al. (1997) | Low | Moderate | Low | Low | Low | Low | Low |
| Licker et al. (2011) | Low | Low | Low | Low | Low | Low | Low |
| Lindenmann et al. (2020) | Moderate | Low | Low | Low | Low | Low | Low |
| Loewen et al. (2007) | Low | Low | High | Moderate | Low | Low | High |
| Mao et al. (2010) | Low | Low | Low | Moderate | Moderate | Low | Moderate |
| Matsuoka et al. (2004) | Low | Low | Moderate | Low | Low | Moderate | Moderate |
| Mazur et al. (2022) | Low | Low | Low | Low | Low | Low | Low |
| Miyazaki et al. (2018) | Low | Low | Low | Low | Low | Low | Low |
| Miyoshi et al. (1987) | Moderate | Low | Low | Low | Low | Low | Low |
| Nagamatsu et al. (2004) | Low | Low | Low | Moderate | Low | Low | Low |
| Olsen et al. (1989) | Low | Low | Low | Low | Moderate | Low | Low |
| Pate et al. (1996) | Low | Low | Low | Low | Moderate | Low | Low |
| Ribas et al. (1998) | Low | Moderate | Low | Low | Low | Low | Low |
| Rodrigues et al. (2016) | Moderate | High | Moderate | Moderate | Moderate | Low | High |
| Shafiek et al. (2016) | Low | Low | Low | Low | Moderate | Low | Low |
| Smith et al. (1984) | Low | Moderate | Low | Low | Low | Low | Low |
| Torchio et al. (1998) | Low | Moderate | Low | Low | Low | Low | Low |
| Torchio et al. (2017) | Low | Moderate | Low | Low | Low | Low | Low |
| Villani et al. (2003) | Low | Low | Low | Moderate | Moderate | Low | Moderate |
| Walsh et al. (1994) | Low | Low | Low | Moderate | Low | Low | Low |
| Wang et al. (2000) | Low | Moderate | Low | Low | Low | Low | Low |
| Win et al. (2005) | Low | Low | Low | Low | Moderate | Low | Low |
| Yakal et al. (2018) | Moderate | Low | Low | Moderate | Low | Low | Moderate |

**Table 2. General study, patient, and CPET characteristics of all included studies.**

| Author | Sample size (n =) | Surgery | | | | | Mean age (years) | Sex | | CPET |
|---|---|---|---|---|---|---|---|---|---|---|
| | | P | L | WR | S | O | | Male | Female | |
| Bechard et al. (1987) | 50 | 10 | 28 | 12 | 0 | 0 | 64 ± 7 | 50 | 0 | Cycle |
| Begum et al. (2016) | 1684 | - | 1403 | - | - | 281 | - | - | - | - |
| Berggren et al. (1984) | 44 | 0 | 44 | 0 | 0 | 0 | 72 | 44 | 0 | Cycle |
| Brat et al. (2016) | 76 | 17 | 47 | 2 | 10 | 0 | 65 ± 6 | 49 | 27 | Cycle |
| Brunelli et al. (2009) | 263 | 27 | 177 | 59 | | 0 | - | - | - | Cycle |
| Brutsche et al. (2000) | 125 | 33 | 77 | 15 | | 0 | 63 ± 11 | 101 | 24 | Cycle |
| Campione et al. (2010) | 99 | 25 | 52 | 0 | 22 | 0 | 67 ± 8 | 80 | 19 | Cycle |
| Chouinard et al. (2022) | 539 | 20 | 471 | 102 | 0 | 0 | 67 ± 7 | 264 | 275 | Cycle |
| Colman et al. (1982) | 59 | - | - | - | - | - | - | - | - | Cycle |
| Epstein et al. (1993) | 42 | - | - | - | - | - | 62 ± 3 | - | - | Cycle |
| Fang et al. (2013) | 107 | 6 | 101 | 0 | 0 | 0 | 65 ± 7 | 104 | 3 | Cycle |
| Fajardo et al. (2014) | 83 | 3 | 63 | 0 | 17 | 0 | 65 ± 10 | 68 | 15 | Cycle |
| Holden et al. (1992) | 16 | 5 | 6 | 2 | 3 | 0 | - | 13 | 3 | Cycle |
| Kasikcioglu et al. (2009) | 49 | 15 | 29 | 3 | 0 | 2 | 61 ± 9 | 44 | 5 | Treadmill |
| Larsen et al. (1997) | 97 | 27 | 52 | 0 | 0 | 18 | 64 ± 9 | - | - | Cycle |
| Licker et al. (2011) | 210 | 39 | - | - | - | - | - | 145 | 65 | Cycle |
| Lindenmann et al. (2020) | 342 | 27 | 315 | 0 | 0 | 0 | 64 ± 9 | 225 | 117 | Cycle |
| Loewen et al. (2007) | 346 | 53 | 286 | 0 | 0 | 7 | - | 245 | 158 | Cycle |
| Mao et al. (2010) | 198 | - | - | - | - | - | - | 163 | 35 | Cycle |
| Matsuoka et al. (2004) | 130 | 0 | 130 | 0 | 0 | 0 | 67 ± 11 | 108 | 22 | Treadmill |
| Mazur et al. (2022) | 294 | 10 | 135 | 140 | 0 | 9 | 66 | 161 | 133 | Cycle |
| Miyazaki et al. (2018) | 209 | - | - | - | - | - | 72 ± 8 | 122 | 87 | Cycle |
| Miyoshi et al. (1987) | 84 | - | - | - | - | - | - | - | - | Cycle |
| Nagamatsu et al. (2004) | 211 | 24 | 187 | 0 | 0 | 0 | 66 ± 8 | 156 | 55 | Cycle |
| Olsen et al. (1989) | 29 | 8 | 13 | 7 | 0 | 1 | 64 ± 5 | 29 | 0 | Cycle |
| Pate et al. (1996) | 12 | 2 | 5 | 2 | 2 | 1 | 64 ± 5 | 10 | 2 | Cycle |
| Ribas et al. (1998) | 65 | 21 | 37 | 7 | 0 | 0 | 66 ± 8 | 64 | 1 | Cycle |
| Rodrigues et al. (2016) | 50 | 11 | 35 | 0 | 4 | 0 | 65 ± 8 | 46 | 4 | Cycle |
| Shafiek et al. (2016) | 51 | 8 | 31 | 0 | 12 | 0 | - | 42 | 9 | Cycle |
| Smith et al. (1984) | 22 | 4 | 12 | 1 | 0 | 5 | 56 ± 2 | 19 | 3 | Cycle |
| Torchio et al. (1998) | 145 | 39 | 106 | 0 | 0 | 0 | 64 | 128 | 17 | Treadmill |
| Torchio et al. (2017) | 263 | 77 | 186 | 0 | 0 | 0 | 65 ± 8 | 212 | 51 | Treadmill |
| Villani et al. (2003) | 150 | 150 | 0 | 0 | 0 | 0 | 57 | 141 | 9 | Cycle |
| Walsh et al. (1994) | 25 | 0 | 11 | 3 | 11 | 0 | 66 ± 7 | 16 | 9 | Cycle |
| Wang et al. (2000) | 57 | 10 | 34 | 4 | 6 | 3 | 64 ± 10 | 39 | 18 | Cycle |
| Win et al. (2005) | 99 | 34 | 62 | 0 | 3 | 0 | 69 ± 8 | 61 | 38 | Treadmill |
| Yakal et al. (2018) | 125 | 16 | 78 | 6 | 0 | 25 | 63 ± 8 | 106 | 19 | Treadmill |

P—pneumonectomy, L—lobectomy, WR—wedge resection, S—segmentectomy, O—other, n—number of patients,

SD—standard deviation, hyphen (-)—missing information

postoperative outcomes [MD: 0.20, 95% CI: 0.08; 0.33] (Fig 8). This was not evident with in-hospital morbidity sub-analysis.

*The relationship between $\dot{V}O_2$ peak as a percentage of predicted and morbidity.* Our pooled analysis showed patients without postoperative complications presented for surgery with a

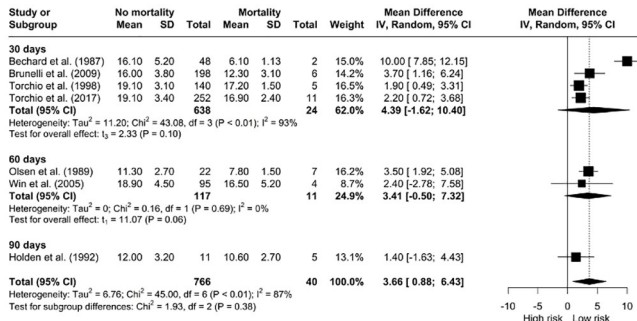

**Fig 2. Meta-analysis of $\dot{V}O_2$ peak (mL/kg/min and L/min) and mortality.**

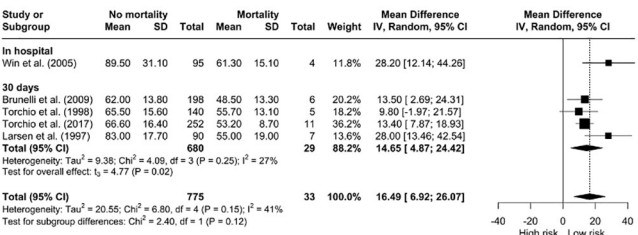

**Fig 3. Meta-analysis of $\dot{V}O_2$ peak as a percentage of predicted and mortality.**

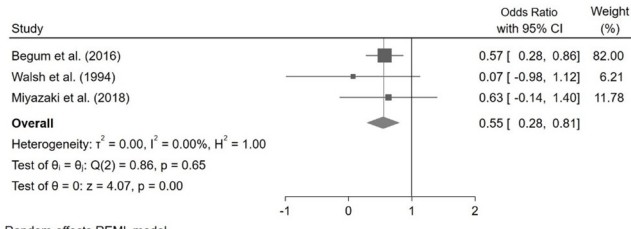

**Fig 4. Pooled odds ratios for mortality when $\dot{V}O_2$ peak > 15mL/kg/min.**

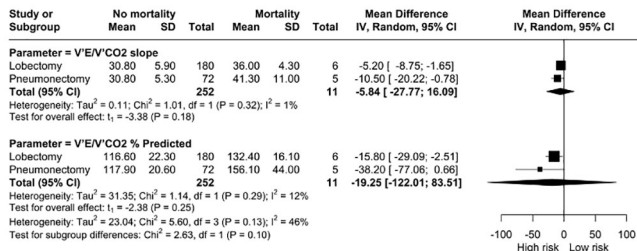

**Fig 5. Meta-analysis of $\dot{V}E/\dot{V}CO_2$ (slope and as a percentage of predicted) and mortality.**

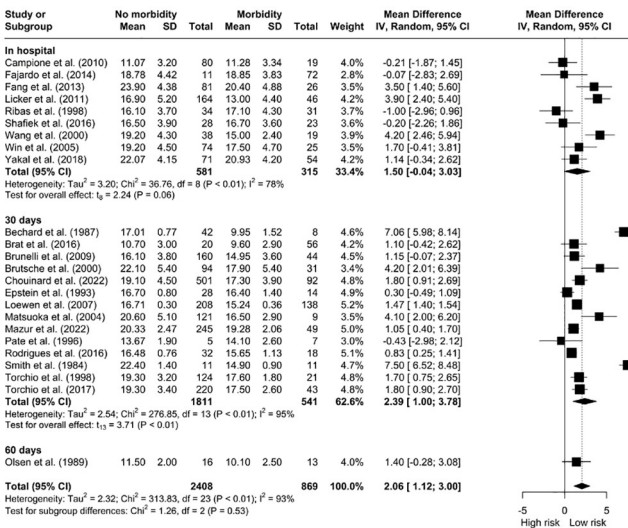

**Fig 6. Meta-analysis of load (watts) and mortality.**

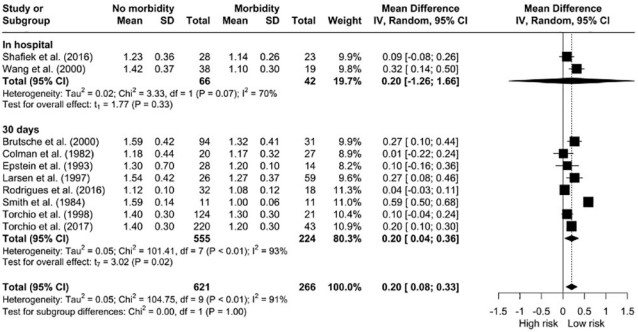

**Fig 7. Meta-analysis of $\dot{V}O_2$ peak (mL/kg/min) and morbidity.**

**Fig 8. Meta-analysis of $\dot{V}O_2$ peak (L/min) and morbidity.**

higher $\dot{V}O_2$ peak as a percentage of predicted [MD: 9.82, 95% CI: 5.88; 13.76] (Fig 9) however, there was significant heterogeneity ($I^2$ = 94%, $X^2$ = 209.27, df = 12, p < 0.01).

*The relationship between $\dot{V}O_2$ peak > 15mL/kg/min and morbidity.* $\dot{V}O_2$ peak > 15mL/kg/min was not associated with fewer complications in pooled data from 3 studies [OR: 0.82, 95% CI: 0.64–1.00] (Fig 10).

*The relationship between $\dot{V}E/\dot{V}CO_2$ and morbidity.* The meta-analysis demonstrated that preoperative estimates of $\dot{V}E/\dot{V}CO_2$ mean differences were significantly lower in patients

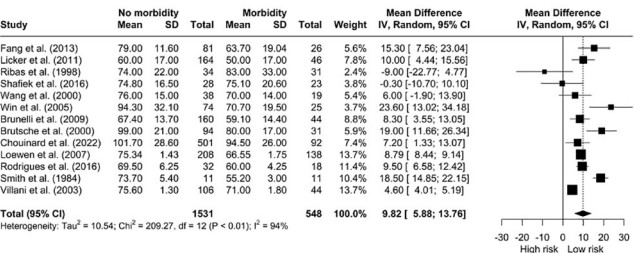

**Fig 9. Meta-analysis of $\dot{V}O_2$ peak as a percentage of predicted and morbidity.**

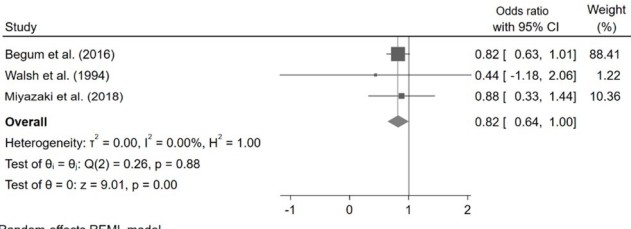

**Fig 10. Pooled odds ratios for morbidity when $\dot{V}O_2$ peak > 15mL/kg/min.**

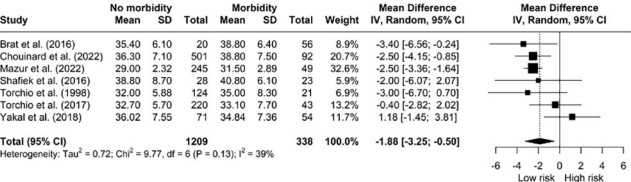

**Fig 11. Meta-analysis of $\dot{V}E/\dot{V}CO_2$ and morbidity.**

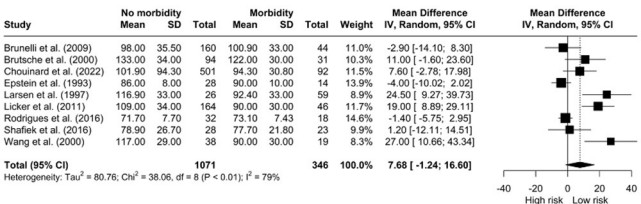

**Fig 12. Meta-analysis of load (watts) and morbidity.**

without complications compared to patients with complications with minimal heterogeneity [MD: -1.88, 95% CI: -3.25; -0.50] (Fig 11).

*The relationship between exercise load and morbidity.* Pooled estimates showed no associations between maximum achieved exercise load (watts) and complication rate [MD: 7.68, 95% CI: -1.24; 16.60] (Fig 12). The studies were significantly heterogenous ($I^2$ = 79%, $X^2$ = 38.06, df = 8, p < 0.01).

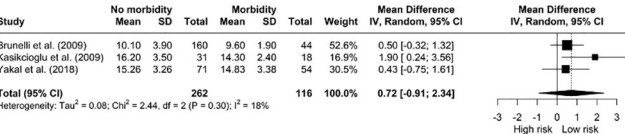

**Fig 13. Meta-analysis of AT (ml/kg/min) and morbidity.**

*The relationship between AT and morbidity*. Our pooled analysis demonstrated no significant difference in mean preoperative AT for patients with or without postoperative complications [MD: 0.72, 95% CI: -0.91; 2.34] (Fig 13). The studies displayed minimal heterogeneity ($I^2$ = 18%, $X^2$ = 2.44, df = 2, p = 0.30).

## Descriptive analysis of outcome measures

Eight studies presented data in categorised variables and different units of measurement and therefore could not be included in the meta-analysis. Nagamatsu et al. were the only authors to comment on $\dot{V}O_2$ peak in mL/body surface area, and Miyoshi et al. were the only authors to comment on $\dot{V}O_2$ at blood lactate of 20mg/dL and both found lower $\dot{V}O_2$ values were associated with increased morbidity and mortality. Other findings included lower risk for mortality in patients who achieved a cycle load > 83 watts, a $\dot{V}E/\dot{V}CO_2$ < 40, and $\dot{V}O_2$ peak as a percentage of predicted $\geq$ 60%. The patients who attained $\dot{V}O_2$ peak as a percentage of predicted $\geq$ 70% and $\dot{V}E/\dot{V}CO_2$ < 40 were less likely to experience postoperative morbidity (Table 3).

## Discussion

This systematic review included 37 studies with 6450 patients. The pooled mean age was 65 years old and majority of patients included in the studies were male. Surgeries performed were predominantly lobectomies. A cycle ergometer was used in most studies to determine CPET parameters. A total of 75% of studies were assessed as having a low RoB. We were able to perform meta-analyses for six CPET variables, namely $\dot{V}O_2$ peak (measured and as a percentage of predicted), $\dot{V}E/\dot{V}CO_2$ (measured and as a percentage of predicted), AT (mL/kg/min) and exercise load (watts).

**Table 3. Summary of findings of papers excluded from the meta-analysis.**

| Author | CPET threshold | | Time Frame | Postoperative outcome | | Estimates, OR, 95% CI |
|---|---|---|---|---|---|---|
| | Favourable | Unfavourable | | Favourable | Unfavourable | |
| Lindenmann et al. (2020) | $\dot{V}O_2$ peak of predicted $\geq$ 60% | $\dot{V}O_2$ peak of predicted < 60% | 10 years from operation | No mortality | Mortality | 0.47 (0.28–0.79) |
| Mao et al. (2010) | $\dot{V}O_2$ peak of predicted $\geq$ 70% | $\dot{V}O_2$ peak of predicted < 70% | In hospital | No cardiopulmonary complications | Cardiopulmonary complications | 0.48 (0.25–0.92) |
| Miyazaki et al. (2018) | $\dot{V}E/\dot{V}CO_2$ < 40 | $\dot{V}E/\dot{V}CO_2$ $\geq$ 40 | In hospital or within 30 days of operation | No cardiopulmonary complications | Cardiopulmonary complications | 0.63 (0.32–1.26) |
| | $\dot{V}E/\dot{V}CO_2$ < 40 | $\dot{V}E/\dot{V}CO_2$ $\geq$ 40 | 90 days from operation | No mortality | Mortality | 0.28 (0.09–0.86) |
| Berggren et al. (1984) | Cycle load > 83 watts | Cycle load < 83 watts | 6 weeks from operation | No mortality | Mortality | 0.29 (0.05–1.80) |

$\dot{V}O_2$ peak is the oxygen consumption at peak exercise and serves as a marker of the body's ability to effectively respond to and adapt to physiological stress. It serves as an important short- and long-term prognostic marker in lung cancer patients, whether undergoing surgery or not [64]. This study found that higher estimates of measured $\dot{V}O_2$ peak were associated with improved survival and better perioperative outcomes. This study also found that $\dot{V}O_2$ peak values of $> 15$mL/kg/min were associated with decreased mortality but not with fewer complications.

$\dot{V}O_2$ peak as a percentage of predicted is the percentage of the patient's measured $\dot{V}O_2$ peak to the expected $\dot{V}O_2$ peak for a healthy patient of the same sex and age [65]. The lower the value, the poorer the functional capacity compared to a matched healthy cohort. Higher estimates of $\dot{V}O_2$ peak as a percentage of predicted were associated with improved survival and fewer complications.

Very little literature exists on the utility of $\dot{V}O_2$ peak as a percentage of predicted for risk stratification. This parameter is important as it does not analyse patient data as an overall measure, but in comparison to patients with purpoted similar profiles. This finding is novel and presents a parameter that has the potential to improve the sensitivity and specificity of CPET as a risk stratification tool. A limitation of $\dot{V}O_2$ peak as a percentage of predicted is that it is based on $\dot{V}O_2$ peak values of healthy matched cohorts from few registries in small populations that may be too population-specific and lack generalisability [66, 67]. The cut-off values presented in the descriptive analysis recommend a threshold of $\geq 60\%$ for decreased mortality and $\geq 70\%$ for decreased complications however, these have not been validated.

A $\dot{V}O_2$ peak value of $< 15$mL/kg/min was found to correlate with increased postoperative pulmonary complications in an earlier systematic review by Benzo et al. [11], this value was also proposed as a tool in the well-known Slinger's "3-legged stool" of preoperative thoracic risk stratification [68] and is recommended in the current guidelines [14]. Our study supports this threshold value for $\dot{V}O_2$ peak for determining patients at high risk of postoperative mortality, however not for postoperative morbidity. Other parameters such as $\dot{V}O_2$ peak as a percentage of predicted seem as significant and need further validation.

Associations between $\dot{V}E/\dot{V}CO_2$ and mortality were not found, however lower estimates of preoperative $\dot{V}E/\dot{V}CO_2$ were associated with a lower complication rate. The value of $\dot{V}E/\dot{V}CO_2$ is the relationship between minute ventilation and carbon dioxide production. It is related to the amount of dead space ventilation and is affected by pulmonary perfusion [69]. In patients with lung cancer, a higher $\dot{V}E/\dot{V}CO_2$ could be found in patients with ventilation-perfusion mismatching due to pulmonary emboli or those with co-existing cardiac disease or pulmonary hypertension. A high $\dot{V}E/\dot{V}CO_2$ should be used to identify patients who would benefit from a preoperative ventilation-perfusion scintigraphy scan or echocardiography to identify any potentially modifiable risk factors for increased cardiopulmonary complications.

The $\dot{V}E/\dot{V}CO_2$ slope is easier to determine than $\dot{V}O_2$ peak as it does not require the patient to reach maximal exercise capacity and is independent of patient volition [70]. It has been reported that $\dot{V}E/\dot{V}CO_2$ cannot be altered through prehabilitation which therefore limits its use [71]. It could, however be used to make decisions in relation to limited surgical approaches. Steffens et al. had previously reported on this parameter in data including liver, lung, oesophageal, bladder and rectal cancer surgeries [72]. This current study focused soley on lung cancer due to its proximity to cardiorespiratory complications. Miyazaki et al. found a $\dot{V}E/\dot{V}CO_2 \geq 40$ to be associated with increased morbidity and mortality in lung cancer patients however threshold values as low as 34 have been used in patients undergoing less

invasive lung surgeries [73]. A receiver-operating-curve (ROC) analysis is still needed to clearly identify a threshold value. This could not be done through this review as raw data was necessary from each study.

Our study did not find a strong association between peak exercise load achieved (watts) and mortality or morbidity. Benzo et al. found a significantly higher peak exercise load achieved in patients without postoperative pulmonary complications [11] and Berggren et al. found increased mortality in patients unable to cycle against > 83 watts. However, the data were not pooled and hence no broad associations could be determined. One would assume that if a patient was able to cycle against a higher resistance, the patient would have better exercise capacity, but this does not seem to translate into changes in perioperative morbidity and mortality.

Previous evidence has shown that a lower AT is associated with higher morbidity and mortality in major abdominal, vascular, transplant and bariatric surgery [4] however we did not find a similar association. The AT is the point above which lactate begins to increase during incremental exercise which indicates increased glycolysis [74]. It is not volitional, unlike $VO_2$ peak [74]. The lack of correlation in this study could potentially be explained by the difference in exercise protocols used. Yakal et al. used the Bruce protocol which is a much steeper, faster ramp than the Naughton protocol used by Kasikcioglu et al. and this could mean patients with severe cardiopulmonary limitations in the study by Yakal et al. could not reach the AT.

The current study recommends that the guidelines incorporate more CPET registries for healthy cohorts in more diverse populations to validate threshold values for $\dot{V}O_2$ peak as a percentage of predicted for different surgical populations. The study supports the use of a $\dot{V}O_2$ peak < 15mL/kg/min for determining patients at high risk of postoperative mortality, however we recommend that the guidelines revise the current threshold $\dot{V}O_2$ peak for determining postoperative morbidiy. Validated threshold values for $\dot{V}O_2$ peak as a percentage of predicted and $\dot{V}E/\dot{V}CO_2$ need to be determined and included in the risk stratification guidelines.

## Strengths and limitations of the study

This study included an up-to-date review of current literature related to CPET and lung resection for lung cancer. In particular, newer evidence has come up since the last systematic review which seems to provide valuable evidence regarding different important CPET parameters. Previously, emphasis was on peak oxygen consumption rather than ventilatory efficiency in association with morbidity and mortality.

Heterogeneity was a limitation in drawing generalised conclusions in this systematic review. There were differences related to the definitions of postoperative complications. Three studies examined all complications (cardiopulmonary and surgical/technical) whereas the remaining studies only examined cardiopulmonary complications. Surgical/technical complications could be explained by surgical expertise, extent of required surgery and postoperative care which could be unrelated to functional capacity and hence undetected by CPET.

Differences were found with regards to blinding of practitioners to the preoperative CPET results. Practitioners in 11 of the included studies reported not using the CPET results to guide decisions on who should undergo surgery or not. While this may be ethically challenging, it does produce more reliable results because it provides a true reflection of the sensitivity and specificity of CPET as a prognostic marker. Differences were also found in the way in which the CPET was performed. Thirty studies used a cycle ergometer whereas six studies used a treadmill. Performing CPET on a treadmill activates more muscle groups, results in a more significant desaturation and produces higher $\dot{V}O_2$ peak than a cycle ergometer [75]. Differences in incremental exercise protocols used, calibration and types of software, registries used

to calculate predicted values and the method of AT determination could have confounded the results. This lack of good-quality standardised evidence could explain why CPET has yet to be motivated for in more centres globally [8].

This study reviewed papers in the English language only. This unfortunately meant excluding approximately ten non-English articles. Translating studies may reduce language bias, but data extraction from translated articles has been shown to be less accurate [76]. This study also included some studies that did not have *a priori* power calculation which could result in underpowered studies.

## Future research

More standardised research on this topic is required. More work is needed to determine the ideal method of prehabilitation to improve postoperative outcomes and to identify any associations between CPET values and quality of life postoperatively. Sensitive and specific thresholds for $\dot{V}O_2$ peak as a percentage of predicted and $\dot{V}E/\dot{V}CO_2$ need to be identified and potentially added to current guidelines. An area of interest could be to define the predictive value of CPET in combination with other risk scores and biomarkers to potentially improve prognostication.

## Conclusion

This study found no association between estimates of $\dot{V}E/\dot{V}CO_2$ and mortality however, lower estimates of $\dot{V}E/\dot{V}CO_2$ were associated with fewer complications. As this measure does not require peak effort, it presents itself as a tool for risk stratification and modification of surgical and intervention strategies. Higher $\dot{V}O_2$ peak estimates, measured and as a percentage of predicted, were associated with both decreased morbidity and mortality. A cutoff value of $\dot{V}O_2$ peak > 15mL/kg/min was consistent with improved survival but not with fewer complications. The literature was significantly heterogenous. Further large scale, prospective, blinded, standardised research is needed to identify and validate threshold values for $\dot{V}E/\dot{V}CO_2$ and $\dot{V}O_2$ peak as a percentage of predicted to improve preoperative risk stratification in patients with lung cancer scheduled for lung resection surgery.

## Supporting information

**S1 Table. PRISMA checklist.**
(PDF)

**S2 Table. Search terms for electronic databases.**
(PDF)

**S3 Table. Mortality data.**
(PDF)

**S4 Table. Morbidity data.**
(PDF)

## Author Contributions

**Conceptualization:** Nabeela Arbee-Kalidas, Hlamatsi Jacob Moutlana, Palesa Motshabi Chakane.

**Data curation:** Nabeela Arbee-Kalidas, Hlamatsi Jacob Moutlana.

**Formal analysis:** Nabeela Arbee-Kalidas, Hlamatsi Jacob Moutlana, Moses Mogakolodi Kebalepile, Palesa Motshabi Chakane.

**Investigation:** Nabeela Arbee-Kalidas, Hlamatsi Jacob Moutlana, Yoshan Moodley, Moses Mogakolodi Kebalepile, Palesa Motshabi Chakane.

**Methodology:** Nabeela Arbee-Kalidas, Hlamatsi Jacob Moutlana, Yoshan Moodley, Moses Mogakolodi Kebalepile, Palesa Motshabi Chakane.

**Project administration:** Nabeela Arbee-Kalidas.

**Supervision:** Hlamatsi Jacob Moutlana, Yoshan Moodley, Moses Mogakolodi Kebalepile, Palesa Motshabi Chakane.

**Writing – original draft:** Nabeela Arbee-Kalidas, Palesa Motshabi Chakane.

**Writing – review & editing:** Nabeela Arbee-Kalidas, Hlamatsi Jacob Moutlana, Yoshan Moodley, Moses Mogakolodi Kebalepile, Palesa Motshabi Chakane.

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
