## [Decision Letter · Decision Letter 0]

28 Jun 2023

PONE-D-23-15371The association between cardiopulmonary exercise testing and postoperative outcomes in patients with lung cancer undergoing lung resection surgery: a systematic review and meta-analysisPLOS ONE

Dear Dr. Arbee-Kalidas,

Thank you for submitting your manuscript to PLOS ONE. After careful consideration, we feel that it has merit but does not fully meet PLOS ONE’s publication criteria as it currently stands. Therefore, we invite you to submit a revised version of the manuscript that addresses the points raised during the review process.

There has only been one review, however there are some very detailed comments on how to improve the article with some consideration for the PRISMA guidelines. Please do take the time to go through them and to respond accordingly. 

We look forward to receiving your revised manuscript.

Kind regards,

Lindsay Bottoms

Academic Editor

PLOS ONE

Journal Requirements:

4. We note that this manuscript is a systematic review or meta-analysis; our author guidelines therefore require that you use PRISMA guidance to help improve reporting quality of this type of study. Please upload copies of the completed PRISMA checklist as Supporting Information with a file name “PRISMA checklist”.

Reviewers' comments:

Reviewer's Responses to Questions

**Comments to the Author**

1. Is the manuscript technically sound, and do the data support the conclusions?

Reviewer #1: Yes

2. Has the statistical analysis been performed appropriately and rigorously? 

Reviewer #1: Yes

3. Have the authors made all data underlying the findings in their manuscript fully available?

Reviewer #1: Yes

4. Is the manuscript presented in an intelligible fashion and written in standard English?

Reviewer #1: Yes

5. Review Comments to the Author

Reviewer #1: Summary

The authors have presented a systematic review and meta-analysis with the aim of investigating potential associations between preoperative CPET parameters and postoperative outcomes in patients undergoing lung resection surgery for lung cancer.

Overall impressions

The study is likely to be of interest to clinicians and researchers. However, the way in which the study was carried out needs a stronger rationale, and greater clarity of some components. The methods are written well with most PRISMA guidelines covered, with minor exceptions (detailed below). The results are interpreted correctly and relate to the objectives of the study. However, the significance of the variables presented is not always clear, and recommendations to reduce the heterogeneity of the literature are not strongly conveyed. Below are my recommendations that should be made in order to improve the quality of this paper, as well as its impact on the field.

Improvements

Major

The rationale for the review could be strengthened. For instance, why are ventilatory equivalents and demarcations of exercise intensity domains of interest to practitioners? A summary of the importance of these factors in the context of existing knowledge could strengthen the rationale. This is a requirement of the PRISMA guidelines.

In the discussion, the implications and explanation of why CPET variables are of use to clinicians is sometimes lacking in appropriate detail. As such, the discussion is quite brief. It would be good to have a more detailed discussion on the utility and significance of these parameters on postoperative outcomes in patients. Further, the heterogeneity of the literature should be discussed in more detail.

Minor

Abstract

Pg 2, Ln 27: 37 should be Thirty-seven.

Pg 2, Ln 27: How many had low bias?

Pg 2, Ln 27: VO2 should have a diacritic mark over the V. Please check throughout.

Pg 2, Ln 41: This sentence structure is difficult to follow, suggest rewording.

Pg 2, Ln 38: how was heterogeneity defined?

Introduction

Pg 4, Ln 48: First two paragraphs could be grouped together.

Pg 4, Ln 58: It may be nice to offer some limitations of the use of CPET to determine fitness for surgery.

Pg 4, Ln 62: Delete “had”

Pg 5, Ln 76: How are you defining “anaerobic threshold”? Is it just oxygen consumption at AT or workload as well?

Pg 5, Ln 80-83: This is too much like your objectives, suggest deletion or modification.

Methods

Pg 5, Ln 92: Full stop missing after reference.

Pg 6, Ln 99-107: Missing information about how studies were grouped for the synthesis.

Pg 7, Ln 118-122: What happened in case of conflicts between authors and how were they settled?

Pg 7, Ln 125-135: Workload is not given here. A few studies use treadmills for CPET, is end speed included or was it excluded from analysis? If excluded, a rationale should be provided.

Pg 8, Ln 150: Outcomes used for grouping could be given here for clarity.

Results

Pg 10, Ln 189: Sex or gender? Be consistent (see below).

Pg 11, Table 2: If using “gender”, the terms “men” and “women” should be used. If “sex”, your current use of male and female is fine. Where information is not given, e.g., the exercise modality for Begum et al., this could be made clearer to the reader.

Pg 15, Ln 275: This is the first time you mention studies being excluded from the MA due to heterogeneity. Was this decided a priori? This should be mentioned in your methods.

Discussion

Ln 16, Pg 290-291: Suggest rewording the sentence beginning “VE/VCO2”, as it doesn’t necessarily reflect the minute ventilation required to eliminate CO2, it’s just the ratio of minute ventilation to carbon dioxide production. This measure is not a definitive indicator of ventilatory inefficiency. This should be expanded on.

Pg 16, Ln 300: Current guidelines should be referenced here.

Pg 17, Ln 304: This sentence could be more specific. Is this end stage exercise load?

Pg 17, 311-313: Reference needed here.

Pg 17, 315: Reference needed here.

Pg 17, 315: This section is surprising, given it is the first time methods other than CPET are discussed. Unsure of the relevance of this.

Pg 18, Pg 329: A more detailed appraisal of the heterogeneity would be welcomed somewhere in the discussion. This is an important finding, and one that doesn’t seem to be explained in sufficient detail.

Conclusion

Pg 19, Ln 354: Although this is a reasonable conclusion to draw, the significance of these variables are not clearly highlighted in the discussion.

6. PLOS authors have the option to publish the peer review history of their article (what does this mean?). If published, this will include your full peer review and any attached files.

Reviewer #1: **Yes: **Ben Hunter

---

## [Author Response · Author response to Decision Letter 0]

12 Aug 2023

We would like to thank you for taking the time to review our paper. Your input was so helpful to strengthen the paper and make it more clinically useful. Please see revised introduction which includes more detail of the existing literature to strengthen the rationale and completely revised discussion section written in more detail including more explanation regarding heterogeneity. All minor revisions attended to. We hope that we have covered all the points sufficiently and look forward to any further feedback.

---

## [Decision Letter · Decision Letter 1]

26 Sep 2023

PONE-D-23-15371R1The association between cardiopulmonary exercise testing and postoperative outcomes in patients with lung cancer undergoing lung resection surgery: a systematic review and meta-analysisPLOS ONE

Dear Dr. Arbee-Kalidas,

Thank you for submitting your manuscript to PLOS ONE. After careful consideration, we feel that it has merit but does not fully meet PLOS ONE’s publication criteria as it currently stands. Therefore, we invite you to submit a revised version of the manuscript that addresses the points raised during the review process.

Thank you for making minor amendments based on the last reviewer's feedback. Could you please address the further minor amendments which have now been suggested. 

We look forward to receiving your revised manuscript.

Kind regards,

Lindsay Bottoms

Academic Editor

PLOS ONE

Journal Requirements:

Reviewers' comments:

Reviewer's Responses to Questions

**Comments to the Author**

1. If the authors have adequately addressed your comments raised in a previous round of review and you feel that this manuscript is now acceptable for publication, you may indicate that here to bypass the “Comments to the Author” section, enter your conflict of interest statement in the “Confidential to Editor” section, and submit your "Accept" recommendation.

Reviewer #1: (No Response)

Reviewer #2: (No Response)

2. Is the manuscript technically sound, and do the data support the conclusions?

Reviewer #1: Yes

Reviewer #2: Yes

3. Has the statistical analysis been performed appropriately and rigorously? 

Reviewer #1: Yes

Reviewer #2: Yes

4. Have the authors made all data underlying the findings in their manuscript fully available?

Reviewer #1: Yes

Reviewer #2: Yes

5. Is the manuscript presented in an intelligible fashion and written in standard English?

Reviewer #1: Yes

Reviewer #2: Yes

6. Review Comments to the Author

Reviewer #1: I would like to thank the authors for their revisions and comments. The paper is improved as a result, and requires very minor amendments in order to be published (see below).

Improvements

Minor

Introduction

Ln 78: Sentence beginning “Historically…” requires a reference.

Ln 95: “< 34” needs units.

Ln 239: I think “peak” has been accidentally deleted? Please check this.

Ln 272: “Figure 54” should be changed to “Figure 5”.

Ln 275: Same as Ln 239.

Ln 363: Unsure of the wording “non-meta-analysed”, but this could just be down to personal preference!

Ln 392: This is an interesting point

Reviewer #2: General Comment

The authors are to be commended for their work on this meta-analysis. I have added a few comments that I hope help improve the manuscript. The authors are free to act on this feedback or contest it if/when they disagree with my comments.

Abstract:

I’d integrate first two sentences of the background section into one (lines 17 – 19), and add definition of VO2 peak (in line 30). I think you can indicate that VO2 peak is (one of the outcomes) that can be derived from CPED, and indeed the measure of CRF.

Line 38 – is this sentence on heterogeneity not already reported in methods?

Line 40 – higher VO2 peak, not better. Then, I find the expression better morbidity and mortality somewhat strange. Better prognosis? Reduced post-operative morbidity or mortality?

Intro

Just a minor point from me. In line 73-75, you highlighted the difference between Benzo et al meta-analysis, and this current work. Would you be able to elaborate on why the observed effect in the general populations (lower VO2peak increases risk complications following pulmonary surgery) may be different in a population with lung cancer?

A minor point in Line 61 – when you state: the potential for incorrect display of graphical data what are you referring to? I have looked at your reference 8 and seem to be unable to locate any further info.

Line 66 and throughout – appreciate AT is still the preferred term, but I would suggest amending the term anaerobic threshold, and refer to gas exchange threshold or lactate threshold instead (see https://physoc.onlinelibrary.wiley.com/doi/epdf/10.1113/JP279963). Also in Line 360-362, that view of AT as the point where aerobic metabolism switches to ‘anaerobic’ metabolism is not generally accepted, and has been contested.

Methods:

Line 115: did you consider whether standard care included exercise recommendations (that is, not a structured exercise programme but the recommendation to stay active?).

Results

As it reads, the numbers don’t quite work in Line 184-5: There were 1096 articles screened. Then - from the titles and abstracts, 1041 were excluded for not meeting the inclusion criteria. A total of 47 studies warranted a full-text review. But 1096 – 1041 is 55, not 47? Please add here that for a further 8 studies full-text article was not found. Also, are these the same 8 studies you describe in lines 189-191? Just needs some clarification.

I think the results may be a bit more clear when it comes to reporting VO2peak (mL/kg/min) – and other physiological parameters, vs reporting the same parameter (e.g. VO2 peak) as a % of predicted. How were these values predicted, as there are many equations that predict for eg. Based on a number of variables. Also, was this prediction done by the authors of the current study, or was it reported in every study?

Line 243 & 274 – these lack of associations/differences may in part be explained by different protocols. A very steep (e.g. 15-20 W per min) may allow participants compared to a shallower protocol, of 5-10 W per min increases. Just consider this in your results and analyses (and interpretation – line 354).

Perhaps the same could be said about AT. There are various protocols to estimate AT (or GET / LT). I am assuming in this context, these estimations are derived from a software (e.g. estimate AT via the v-slope method), but there is likely some variability and error in estimates, and differences between labs, equipment, protocols, etc. that may confound the results you are reporting herein.

Discussion

Perhaps the authors could consider adding CP as a (plausible marker).

I feel the discussion remains too generic, and there are some missed opportunities. Please do try and direct the discussion a bit more towards your own results. For example, in the second / third paragraph of the discussion, you report the threshold of 15 mL for a VO2peak – is that value still applicable considering the results reported in this meta-analysis? Should be revisited? The discussion reads a bit like a review/introduction. Also – line 326/7 – clearly indicate here again what those cutoff values are.

I think you should emphasise, more clearly, what the results of this meta-analysis. For example, the current guidelines was for VO2 peak to reach 15 mL, is that still the case after this study?

7. PLOS authors have the option to publish the peer review history of their article (what does this mean?). If published, this will include your full peer review and any attached files.

Reviewer #1: **Yes: **Ben Hunter

Reviewer #2: No

---

## [Author Response · Author response to Decision Letter 1]

4 Nov 2023

Thank you to the two reviewers for their time and expertise. We appreciate the thorough feedback.

---

## [Decision Letter · Decision Letter 2]

22 Nov 2023

The association between cardiopulmonary exercise testing and postoperative outcomes in patients with lung cancer undergoing lung resection surgery: a systematic review and meta-analysis

PONE-D-23-15371R2

Dear Dr. Arbee-Kalidas,

We’re pleased to inform you that your manuscript has been judged scientifically suitable for publication and will be formally accepted for publication once it meets all outstanding technical requirements.

Kind regards,

Lindsay Bottoms

Academic Editor

PLOS ONE

Additional Editor Comments (optional):

Reviewers' comments:

Reviewer's Responses to Questions

**Comments to the Author**

1. If the authors have adequately addressed your comments raised in a previous round of review and you feel that this manuscript is now acceptable for publication, you may indicate that here to bypass the “Comments to the Author” section, enter your conflict of interest statement in the “Confidential to Editor” section, and submit your "Accept" recommendation.

Reviewer #1: All comments have been addressed

Reviewer #2: (No Response)

2. Is the manuscript technically sound, and do the data support the conclusions?

Reviewer #1: Yes

Reviewer #2: Yes

3. Has the statistical analysis been performed appropriately and rigorously? 

Reviewer #1: Yes

Reviewer #2: Yes

4. Have the authors made all data underlying the findings in their manuscript fully available?

Reviewer #1: Yes

Reviewer #2: Yes

5. Is the manuscript presented in an intelligible fashion and written in standard English?

Reviewer #1: Yes

Reviewer #2: Yes

6. Review Comments to the Author

Reviewer #1: Thank you for addressing the points made by myself and the other reviewer. The other reviewer has suggest the addition of critical power (CP) as a plausible marker. However, I'm not sure whether this would be additive and seems separate to the focus of the MA which is CPET.

Reviewer #2: General comments:

Thanks to the authors for the newer version of the manuscript. I am not sure I can identify a document where I can see reviewer’s comments and then the response from the authors to each comment.

I can see some of the comments have not been addressed. I do not think this is an issue but it would have been useful to see the thinking behind the authors when rejecting some of our comments.

However, having read the document again I have a few minor final suggestions.

Abstract

Line 34, but also throughout – is it worth stating that “complications” refers to postoperative complications, even at risk of stating the obvious?

Conclusion – first sentence may want to refer to the specific population of interest.

Line 91 – when you state the predicted VO2 peak, perhaps you can explain how this value was predicted (what equation was used)? You refer to this in line 347-348 but this may need to be addressed before?

Line 188-200 what caused the changes in the numbers of study screened and studies that were then included in the final analysis? (compared to what was originally reported).

Not sure you have done this (or whether you can with the data that you have), but would it be possible to combine the %predicted and actual VO2peak to assess risk?

7. PLOS authors have the option to publish the peer review history of their article (what does this mean?). If published, this will include your full peer review and any attached files.

Reviewer #1: No

Reviewer #2: No

---

## [Editor Report · Acceptance letter]

28 Nov 2023

PONE-D-23-15371R2 

The association between cardiopulmonary exercise testing and postoperative outcomes in patients with lung cancer undergoing lung resection surgery: a systematic review and meta-analysis 

Dear Dr. Arbee-Kalidas:

I'm pleased to inform you that your manuscript has been deemed suitable for publication in PLOS ONE. Congratulations! Your manuscript is now with our production department. 

Kind regards, 

on behalf of

Dr. Lindsay Bottoms 

Academic Editor

PLOS ONE